# MK-2206 Alleviates Renal Fibrosis by Suppressing the Akt/mTOR Signaling Pathway In Vivo and In Vitro

**DOI:** 10.3390/cells11213505

**Published:** 2022-11-05

**Authors:** Meiling Chen, Yihang Yu, Tao Mi, Qitong Guo, Bin Xiang, Xiaomao Tian, Liming Jin, Chunlan Long, Lianju Shen, Xing Liu, Jianbo Pan, Yuanyuan Zhang, Tao Xu, Deying Zhang, Guanghui Wei

**Affiliations:** 1Department of Urology, Children’s Hospital of Chongqing Medical University, Ministry of Education Key Laboratory of Child Development and Disorders, Chongqing Key Laboratory of Pediatrics, National Clinical Research Center for Child Health and Disorders, China International Science and Technology Cooperation Base of Child Development and Critical Disorders, Chongqing Key Laboratory of Children Urogenital Development and Tissue Engineering, Chongqing 400014, China; 2Center for Novel Target and Therapeutic Intervention, Institute of Life Sciences, Chongqing Medical University, Chongqing 400016, China; 3Wake Forest Institute for Regenerative Medicine, Wake Forest School of Medicine, Winston-Salem, NC 27101, USA; 4Biomanufacturing Center, Department of Mechanical Engineering, Tsinghua University, Beijing 100084, China

**Keywords:** chronic kidney diseases, renal fibrosis, MK-2206, Akt/mTOR signaling pathway

## Abstract

Renal fibrosis is a common pathological feature of various kidney diseases, leading to irreversible renal failure and end-stage renal disease. However, there are still no effective treatments to reverse renal fibrosis. This study aimed to explore the potential mechanism of a targeted drug for fibrosis. Here, unilateral ureteral obstruction (UUO)-treated mice and a TGF-β1-treated human renal tubular epithelial cell line (HK-2 cells) were used as models of renal fibrosis. Based on the changes of mRNA in UUO kidneys detected by transcriptome sequencing, MK-2206, an Akt inhibitor, was predicted as a potential drug to alleviate renal fibrosis through bioinformatics. We dissolved UUO mice with MK-2206 by gastric gavage and cultured TGF-β-induced HK-2 cells with MK-2206. Histopathological examinations were performed after MK-2206 intervention, and the degree of renal fibrosis, as well as the expression of Akt/mTOR pathway-related proteins, were evaluated by immunohistochemical staining, immunofluorescence staining, and Western blot. The results showed that MK-2206 significantly improved the pathological structure of the kidney. Furthermore, MK-2206 intervention effectively inhibited UUO- and TGF-β1-induced epithelial-mesenchymal transition, fibroblast activation, and extracellular matrix deposition. Mechanistically, MK-2206 treatment attenuated the activation of the Akt/mTOR signaling pathway. Taken together, our study revealed for the first time that MK-2206 is a promising drug for the improvement of renal fibrosis.

## 1. Introduction

In recent years, kidney disease has become a global public health problem that seriously affects the quality of patients’ life. According to the statistics, the global prevalence of chronic kidney diseases (CKDs) was 9.1% in 2017, with a total of 697 million patients worldwide [1]. CKD rose from 27th in 1990 to 18th in 2010 on the list of causes of total deaths worldwide, and CKD is the third highest cause of premature mortality, behind AIDS and diabetes mellitus [2].

Renal fibrosis is a characteristic change in the pathogenesis of CKD and is a common pathological outcome in various progressive renal diseases [3]. Moreover, fibrosis is the ultimate pathological process of maladaptive repair, and the main pathological change is the excessive accumulation of activated and amplified extracellular matrix (ECM). The activation of fibrosis can be triggered by various stimuli, including infections, chemicals, metabolic disorders, autoimmune reactions, and mechanical injury [4]. As part of the basic process of wound healing, fibrosis may be beneficial; however, if the initial fibrotic process is not fully resolved, it is likely to be reactivated later in life by the activation of hypertension, diabetes mellitus, or the immune system, subsequently leading to the massive deposition of ECM and the replacement of normal tissue with permanent scar tissue [5,6]. In the kidney, massive ECM deposition leads to fibrous scar formation and renal tissue remodeling, ultimately resulting in a reduction of intact renal units and loss of renal function. Renal fibrosis is characterized by progressive tissue scarring, which leads to glomerulosclerosis and tubulointerstitial fibrosis [5]. However, renal fibrosis is not just a static “scar”, but a dynamic process involving complex cellular changes. It involves numerous signaling pathways, such as TGF-β/SMAD, Wnt/β-catenin, and the c-Jun amino-terminal kinase (JNK) signaling pathways [7]. It has also been suggested that the Akt/mTOR pathway is closely related to fibrosis [8]. There are several key steps in the formation of renal fibrosis, such as immune cell infiltration, epithelial–mesenchymal transition (EMT), fibroblast activation, apoptosis, and matrix production/degradation imbalance [9]. Due to the strong association between fibrosis and renal disease and the fact that in experimental models, renal fibrosis can easily be detected and quantified, fibrosis is always regarded as an important way to investigate potential therapeutic targets known as “antifibrosis”. It is worth noting that several steps, such as EMT, fibroblast activation, and matrix production/degradation imbalance, cannot be ignored when studying the mechanisms of antifibrosis.

Currently, the standard clinical treatment used to slow CKD progression is to block the renin–angiotensin–aldosterone system (RAAS) using angiotensin-converting enzyme inhibitors (ACEI), angiotensin II receptor type 1 (AT1) antagonists, or direct renin blockers. However, these drugs mainly play an indirect antifibrotic role, and blocking the RAAS has a limited effect on inhibiting the progression of CKD [10]. A variety of drugs that directly target renal fibrosis have been tested in clinical trials, such as fresolimumab (a human monoclonal antibody anti-TGF-β) and STX100 (a humanized anti-α5β6 integrin antibody) [10,11], but none of these drugs can be used in the clinic yet due to safety and efficacy concerns. In conclusion, there are no sufficiently effective drugs that specifically target renal fibrosis in the clinic, which happens to be a great obstacle to the therapy of CKD.

Here, we successfully constructed a unilateral ureteral obstruction (UUO) renal fibrosis model, identified the key differentially expressed genes using high-throughput mRNA sequencing, and then combined the key differential genes and CMap database to predict a potential drug (MK-2206, an Akt inhibitor) targeting renal fibrosis. Finally, we validated the key effects and specific mechanisms of the drug in anti-nephrogenic fibrosis in vivo and in vitro.

## 2. Materials and Methods

### 2.1. Ethics Statement

The study concerning animals followed the guidelines of the Helsinki Declaration and was authorized by the Research Ethical Committee of the Children’s Hospital of Chongqing Medical University. Eighteen C57BL/6 male mice of 8–10 weeks (weighing 18–25 g) were obtained from the Experimental Animal Center of Chongqing Medical University (SPF, License No. SYXK (Chongqing) 2007-0001) and housed in the Experimental Animal Center of Chongqing Medical University Children’s Hospital (SPF, License No. SYXK (Chongqing) 2007-0016). Additionally, the mice were raised in polycarbonate cages under an illumination schedule of semidiurnal light/semidiurnal dark, and they were provided with free access to food and water. The HK-2 cells (an adult renal proximal tubular epithelial cell line) were obtained from the China Center for Type Culture Collection (CCTCC).

### 2.2. Animal Experiments (UUO Mouse Model and Intervention with MK-2206)

The mice were randomly divided into three groups (n = 6 in each group): (1) sham mice; (2) UUO mice; and (3) MU (UUO+MK-2206) (120 mg/kg) mice. We treated the mice in each group as follows: the UUO mice and UUO+MK-2206 mice were anesthetized with isoflurane. An incision was made in the left epigastrium to expose the left kidney and ureter, and then the ureter was firmly ligated with two 4-0 silk wires in the upper third of the ureter. To prevent the occurrence of retrograde urinary tract infection, we incised the ureter with scissors between the two ligation points. As a control, the sham group were operated on in the same way except for ureteral ligation. From the day of surgery, the MK-2206 mice were administered the corresponding dose (120 mg/kg) of MK-2206 (MK-2206 dihydrochloride, MCE, NJ, USA) by oral gavage on alternate days, with a total of four gastric gavages (0, 2, 4, and 6 days). The other two groups were administered the corresponding volume of saline by oral gavage on alternate days for a total of four times. The mice were sacrificed on day 7 after surgery and their blood was collected for biochemical analysis. Subsequently, a portion of each left kidney was immersed in 4% paraformaldehyde and the other portion was stored in a −80 °C refrigerator for subsequent experiments.

### 2.3. Serum Biochemical Measurements

The biochemical parameters (including creatinine and urea nitrogen, Scr, and BUN) in the serum of each group were examined. After the mice were anesthetized with 10% chloralhydrate, their blood samples were collected and centrifuged at 4 °C for 10 min to obtain the serum, which was then sent to the Laboratory Department of the Children’s Hospital of Chongqing Medical University for the measurement of the serum’s biochemical parameters (Scr and BUN).

### 2.4. Transcriptome Sequencing and Results Analysis

Six different kidney tissue specimens in the sham group and UUO group (n = 3 in each group) were collected for transcriptome analysis, and the high-throughput sequencing part was done by BGI (BGI Tech, Shenzhen, China). The details are as follows: in accordance with manual instructions, Trizol (Invitrogen, Carlsbad, CA, USA) was used to extract the total RNA from the tissues. Moreover, the whole RNA was conducted qualitatively and quantificationally through a NanoDrop and Agilent 2100 bioanalyzer (Thermo Fisher Scientific, Waltham, MA, USA). The single-strand circle DNA (ssCir DNA) was formatted as the final library. Then, the final library was amplified with phi29 to make a DNA nanoball (DNB) and sequenced on a BGIseq500 platform (BGI, Shenzhen, China).

Differentially expressed genes (DEGs) between the UUO and sham groups were analyzed through the R software (R4.0.1) edgeR package, with a *p*-value < 0.05 and |logFC| > 2 as screening conditions, and genes with low expression abundance were removed. Next, the clusterProfiler package was used to perform gene ontology (GO) and Kyoto Encyclopedia of Genes and Genomes (KEGG) enrichment analyses [12]. Meanwhile, we used Gene Set Enrichment Analysis (GSEA) to identify any potential pathways, and *p* < 0.05 was considered statistically significant.

### 2.5. Identification of Key DEGs

To further identify key DEGs, the above DEGs were uploaded to the STRING database [13] to build protein–protein interaction (PPI) networks. After that, Cytoscape software was used for visualization [14] and key DEGs were identified by the MCC algorithm. The top 150 up- and downregulated DEGs were selected as a key set of genes for downstream analysis.

### 2.6. Drug Prediction

The connectivity map (CMap) is a drug-screening tool that stores gene expression data from a wide range of cells (about 22,000 genes). It is most notably a query function for assessing the similarity between the up- and downregulated genes provided by the researcher and the expression profile information of more than 1300 small-molecule drug treatments, which are included in the database “build2”. It is also an effective genome-based tool to discover new chemopreventive drugs. Potential antifibrotic drugs were predicted using the aforementioned differential genes. Subsequently, the 3D structure of the small-molecule drug was visualized using the PubChem database (https://pubchem.ncbi.nlm.nih.gov/, accessed on 28 July 2022).

### 2.7. Histological Examination

To observe the histomorphology of the mouse kidneys, hematoxylin and eosin (H&E) and Masson’s trichrome were used to stain the kidney tissues following the standard staining protocols [15].

### 2.8. Cell Culture

The in vitro study included three groups: (1) the control group, (2) the TGF-β1 group, and (3) the MT (TGF-β1+MK-2206) group. The details are as follows: we cultured the HK-2 cells in Dulbecco’s modified Eagle’s medium nutrient mixture F-12 (DMEM/F12, Gibco, NY, USA) containing 10% fetal bovine serum (FBS, Corning, NY, USA) and 1% penicillin/streptomycin in an environment with 5% CO_2_ at 37 °C, and changed the culture medium every 3 days. After digesting the cells with trypsin (Sigma, MO, USA), the cells were evenly seeded on 24/6-well plates at a density of 2.0 × 10^4^/1.0 × 10^5^ cells per well. The medium was changed and MK-2206 (MCE, NJ, USA) was added at the indicated concentrations (24-well plates, 0, 1 μM/well; 6-well plates, 0, 0.5, 1, 2.5, and 5 μM/well) until the cells were plastered. Six hours later, recombinant human transforming growth factor β1 (TGF-β1) (MCE, NJ, USA) (10 ng/mL) was used to stimulate the cells. After 48 h of co-culture, the cell slides in the 24-well plates were used for immunofluorescence staining, and the RNA extracted from cells in the 6-well plates was collected for real-time quantitative polymerase chain reaction (RT-qPCR). 

### 2.9. Immunohistochemistry Staining

In the immunohistochemistry staining of the kidney paraffin sections, xylene was used to deparaffinize them and ethanol in gradually decreasing concentrations was used to dehydrate them. Then, microwave heating was used to repair the antigen of the sectioned tissues, and 3% H_2_O_2_ was used to quench the endogenous peroxidases, followed by washing the sections with phosphate buffered solution (PBS). Following this, the slides were blocked with 0.1% Triton X-100 (Solarbio, Beijing, China) and 0.5% bovine serum albumin (Solarbio, Beijing, China) for 1 h, and then incubated with primary antibodies phospho-Akt (p-Akt) (1:200, Proteintech, Wuhan, China) and phospho-mTOR (p-mTOR) (1:200, Proteintech, Wuhan, China). Next, the secondary antibody of the corresponding species (1:200, Zhongshan, Beijing, China) was incubated for 1 h. Then, 3,3′-diamnobenzidine (DAB, Abcam, Cambridge, UK) was used as the positive coloring reagent, and hematoxylin was used as the counterstain. Finally, all specimens were photographed with an optical microscope (Olympus, TKY, Japan).

### 2.10. Immunofluorescence Staining

The cell-culturing method was referred to above. A solution of paraformaldehyde was used to fix the cells for 20 min, followed by permeabilizing with 0.1% Triton X-100 (Solarbio, Beijing, China) for 15 min and blocking with 0.5% bovine serum albumin (Solarbio, Beijing, China) for 1 h. Subsequently, the primary antibodies Vimentin (1:200, ZEN-BIOSCIENCE, Chengdu, China), E-cadherin (1:200, Proteintech, Wuhan, China), α-SMA (1:200, ZEN-BIOSCIENCE, Chengdu, China), p-Akt (1:200, Proteintech, Wuhan, China), and p-mTOR (1:200, Proteintech, Wuhan, China) were incubated with the cells for 12 h at 4 °C. The next day, the cells were incubated with a corresponding fluorescent secondary antibody (1:200, Proteintech, Wuhan, China) for 1 h. Finally, the specimens were incubated with DAPI (1:200, Beyotime, Shanghai, China) for 1 h in order to stain the nucleus. Fluorescence microscopes (K10587, Nikon, TKY, Japan) were used to take all of the pictures.

The kidney paraffin sections were obtained as described above and were treated in the same way as the immunohistochemical stain until being incubated with primary antibodies. Then, the slides were treated in the same way as for the cell immunofluorescence staining for incubation with primary antibodies and the remaining steps.

### 2.11. Western Blot 

The HK-2 cells treated by the above method and the kidney tissues stored at −80 °C were lysed in RIPA Lysis Buffer (Beyotime Technology, Shanghai, China) including 1% phenylmethanesulfonyl fluoride (MCE, NJ, USA). They were then exposed to ultrasonic pulverization at 4 °C for 10 min and centrifugation at 12,000 rpm for 20 min in a 4 °C centrifuge. Each protein sample was mixed with 5× buffer in a 4:1 ratio and boiled for 10 min to obtain the required protein samples. We subjected equal amounts of proteins to 7.5–12.5% SDS-PAGE (EpiZyme, Shanghai, China) and then transferred them to polyvinylidene fluoride membranes (Millipore, MA, USA). The membranes were blocked with blot-blocking buffer (NCM, Suzhou, China) for 10 min. The immunoblotting was performed with antibodies corresponding to the following proteins: anti-E-cadherin, anti-Fibronectin, anti-Akt, anti-mTOR and anti-p-mTOR (1:1000, Proteintech, Wuhan, China), anti-Vimentin, anti-α-SMA, anti-p-Akt and anti-GAPDG (1:1000, ZEN-BIOSCIENCE, Chengdu, China), and anti-Collagen I (1:1000, Bioss, Beijing, China) for 12 h at 4 °C. The membranes were washed in TBST three times and incubated with the secondary antibodies obtained from Zhongshan (Beijing, China). The enhanced chemiluminescence substrate (34075, Thermo Scientific, Waltham, MA, USA) was used to visualize the protein bands and the chemiluminescent reagent ECL (ECL, Bio-Rad, CA, USA) was used to capture the images. The Western blot experiments were repeated independently at least three times.

### 2.12. Real-Time PCR

The total RNAs were extracted from the renal tissues by TRIzol reagent (Invitrogen, Carlsbad, CA, USA) and from HK-2 cells using Simply P Total RNA Extraction Kits (BioFlux, Beijing, China) under the guidance of the manufacturer’s instructions. We made 1 μg of total RNA reverse transcribed through RT Master Mix for qPCR (gDNA digester plus) (MCEs, NJ, USA), and the generated cDNA was used as a template for the PCR. The mRNA level of the genes was detected by SYBR Green Master Mix. All primer sequences are listed in Table 1. GAPDH was used as an internal reference and the genes were quantified relatively using the comparative Ct method formula (2−ΔΔCt).

### 2.13. Statistical Analysis

All data are presented as the mean ± standard deviation (SD) and all of the experiments were conducted at least three times. Student’s *t*-tests were used to compare the statistical differences between two groups, and one-way ANOVA was used for more than two groups. Additionally, the data were calculated with GraphPad Prism (GraphPad Software, version 8.0.2, CA, USA). A *p*-value less than 0.05 was considered statistically significant.

## 3. Results

### 3.1. Establishment of the Renal Fibrosis Model

In the current study, an obstructive kidney model was constructed by UUO in mice. By measuring the levels of Scr and BUN, we found that the renal function of mice in the UUO kidneys was significantly lower compared to the normal kidneys (Figure 1H,I). H&E staining assessed the structural damage of the kidney and Masson’s trichrome staining revealed the degree of fibrosis (Figure 1A). In the sham group, the glomeruli and tubules were morphologically normal and structurally intact, with only a small amount of interstitial collagen fibers; on the contrary, the kidney showed obvious structural disorders in the UUO mice, including compensatory tubular dilation and atrophy, massive infiltration of inflammatory cells, and a large number of collagen fiber streaks in the renal interstitium, indicating significant renal injury and fibrosis. Western blot analysis was used to further investigate the extent of the renal fibrosis, and the results suggested that the expression of E-cadherin, the epithelial cell marker, was significantly reduced in the obstructed kidneys compared with normal kidneys. However, the expression of the myofibroblast markers α-SMA and Vimentin was remarkably increased, and the interstitial Collagen I and Fibronectin were also expressed at a higher level (Figure 1B–G). In all, these results suggest that the UUO model successfully induced fibrosis in the kidney.

### 3.2. PI3K/Akt Signaling Pathway Was Enriched in the UUO Model

In order to investigate the possible mechanisms of renal fibrosis, we conducted differential analysis as well as enrichment analysis of the renal transcriptome sequencing results from the sham and UUO groups. Through the differential analysis, we identified 1994 genes that were aberrantly expressed in the UUO model. Among them, 1434 genes were upregulated and 560 genes were downregulated (Figure 2A,B). Based on the differential genes, we performed GO and KEGG enrichment analysis (Figure 2C,D) and found that the differential genes were associated with cellular expression. We found that the differential genes were associated with the extracellular matrix, which demonstrated that the UUO model did cause fibrosis-related changes. For the pathways, we mainly enriched the PI3K/Akt, PPAR, and other signaling pathways. Moreover, the GSEA enrichment analysis revealed the activation of the PI3K/Akt signaling pathway in the UUO model (Figure 2G–L). Further, we used Western blot and immunofluorescence to confirm this result.

### 3.3. CMap Predicts the Akt Inhibitor MK-2206 as a Pivotal Targeted Drug

The String database combined with Cytoscape software was used to find the most critical genes among the upregulated as well as downregulated genes (Figure 2E,F). Among the upregulated genes, *ASPM*, *TPX2*, and other genes were identified as hub genes. Among the downregulated genes, *ACOX1*, *EHHADH*, and others were the most critical genes. It was reported that the downregulation of *ACOX1* effectively inhibited the fatty acid β-oxidation pathway, thereby decreasing kidney function and possibly promoting the fibrotic process in mouse kidneys [16]. The knockdown of *EHHADH* led to the downregulation of fatty acid metabolic processes and the upregulation of inflammation, as well as proximal tubule cell injury, providing an important condition for the development of renal fibrosis [17,18]. Drug prediction was performed based on these key genes. We predicted that MK-2206, Voapaxar, etc., might be useful for the treatment of UUO-induced fibrosis (Figure 2M). MK-2206, a potent Akt inhibitor, is consistent with our study of PI3K/Akt signaling pathway activation in the UUO model. Therefore, we selected MK-2206 and validated its effects in vitro and in vivo. 

### 3.4. MK-2206 Improves the Pathological Structure and Suppresses the Inflammatory Reaction in the Kidney of UUO Mice

To illustrate how MK-2206 protected against renal injury, we conducted Masson trichrome and H&E staining. As a result, it was revealed that the extent of tubular injury was smaller and the severity of injury was less in the MU kidney compared with the UUO kidney. Meanwhile, the collagen fibers in the tubular interstitium were significantly reduced (Figure 3A). Fibrosis is usually caused by chronic inflammation lasting for months, during which the inflammation, tissue remodeling, and repair processes occur simultaneously [19]. After the kidney is damaged by multiple factors, a variety of inflammatory cells (macrophages, lymphocytes, etc.) are activated and raised to the site of injury to take part in the immune response. Therefore, inflammation is the primary cause of fibrosis. These inflammatory cells release transforming growth factor β1 (TGF-β1), which causes fibroblast activation and promotes the formation of renal fibrosis [20]. In order to determine whether MK-2206 has anti-inflammatory properties, we detected the levels of IL-1β, TGF-β1, and IL-6 by qPCR. We found that the mRNA expression of TGF-β1, IL-1β, and IL-6 was remarkably increased in the UUO kidneys (Figure 3B–D), and the expression of inflammatory factors were remarkably reduced after MK-2206 intervention.

### 3.5. MK-2206 Inhibits Renal Fibrosis in UUO Model

During epithelial mesenchymal transition, renal tubular epithelial cells lose their own epithelial cell phenotype and acquire a mesenchymal cell phenotype, ultimately leading to renal fibrosis [21]. To determine that MK-2206 inhibited the development of EMT in UUO kidney, this study examined the level of E-cadherin (a marker of renal tubular epithelial cell) by immunofluorescence staining and Western blot. It was indicated that the UUO group showed a significant downregulation of E-cadherin expression compared to the sham group, but MK-2206 reversed this result (Figure 4A,C). 

α-SMA is a component of the actin cytoskeleton. It is recognized as a myofibroblast marker, and is also used to identify renal epithelial EMT [22]. Both Western blot assays and immunofluorescence staining indicated that the expression of myofibroblast markers (α-SMA, Vimentin) was remarkably upregulated in UUO (Figure 5A–C), but this result was reversed by MK-2206. The transcriptional factors associated with EMT (Twist and Snail) were also analyzed in different kidneys by Western blot and the results suggested the same: that the intervention of MK-2206 effectively suppressed the UUO-induced high expression of the two factors (Figure 4C).

Apart from myofibroblast formation, there is also a crucial step in renal fibrosis, namely, the amassment of extracellular matrix molecules (ECM). ECM are derived from activated fibroblasts or myofibroblasts and consist of various ingredients (Collagen I, Fibronectin etc.). Excessive ECM deposition is the result of an impaired balance between matrix protein generation and degradation [23]. In the present study, Fibronectin and Collagen I were quantified through Western blot, and according to the results, they were markedly more expressed in UUO kidney compared with normal kidney. However, MK-2206 inhibited their expression (Figure 5G).

### 3.6. MK-2206 Inhibits the Activation of Akt/mTOR Pathway in the UUO Model

To verify the ability of MK-2206 to attenuate renal fibrosis and its specific mechanism of action, we examined the phosphorylation levels of related proteins in the Akt-mTOR pathway. The results of immunohistochemistry and immunofluorescence staining revealed that UUO kidneys overexpressed phosphorylated Akt and mTOR compared to the normal kidneys, while MK-2206 effectively inhibited the phosphorylation of Akt and mTOR (Figure 6A,B,G). The same results were obtained by quantification of Western blot, where the phosphorylation of Akt and mTOR was activated in UUO kidneys, while it was effectively suppressed by the intervention of MK-2206 (Figure 6C,H).

### 3.7. MK-2206 Inhibits TGF-β1-Induced Fibrosis in HK-2 Cells

Based on the validation of the antifibrotic effect of MK-2206 in vivo, we investigated the suppression effect of MK-2206 on TGF-β1-induced renal fibrosis in vitro. Referring to previous studies [24,25,26], we interfered HK-2 cells induced by TGF-β1 (10 ng/mL, 48 h) with different concentrations (0, 0.5, 1, 2.5, and 5 μM) of MK-2206. qPCR was used to detect the mRNA expression of Collagen I and Fibronectin, and the results are shown in Figure 3E,F. 1 μM concentration of MK-2206 effectively reduced the mRNA levels of Collagen I and Fibronectin in TGF-β1-induced HK-2 cells. Based on this result, we chose 1 μM of MK-2206 for the subsequent experiments.

TGF-1 plays a significant role in myofibroblast differentiation during the fibrosis process [27], and it is commonly used to construct fibrosis models with HK-2 cells in vitro [15,26]. The expression of Vimentin, α-SMA, and E-cadherin was detected by immunofluorescence staining and Western blot, both showing that MK-2206 effectively restored the E-cadherin expression in HK-2 cells after TGF-β1 induction (Figure 4B,D) and decreased the α-SMA and Vimentin expression (Figure 5D–F). The Western blot detection of the ECM-related proteins, Collagen I and Fibronectin, showed that MK-2206 significantly inhibited their upregulation after TGF-β1 induction (Figure 5H).

### 3.8. MK-2206 Inhibited the Activation of Akt-mTOR Signaling Pathway in HK-2 Cells

We observed changes in proteins involved in the Akt-mTOR pathway to further verify the molecular mechanism of MK-2206 in TGF-β1-treated HK-2 cells. The results suggested that MK-2206 distinctly inhibited the TGF-β1-induced phosphorylation of Akt, mTOR (Figure 6D–F,I).

## 4. Discussion

The fibrosis of the kidney is the most common cascade reaction in chronic kidney disease and a key pathological factor in acute kidney injury (AKI), CKD, and ESRD [28]. In the end stage of renal disease, dialysis or renal transplantation are the only options due to irreversible tissue damage and functional impairment. However, such treatments are far from meeting the actual demand due to the economic level and the constraints of kidney sources [29]. Thus, to prevent or slow the progression of CKD and end-stage renal disease, the effective prevention of interstitial fibrosis is essential. In our study, we visualized changes at the transcriptional level in UUO kidneys by transcriptome sequencing and predicted possible drugs to alleviate renal fibrosis using bioinformatics analysis. Combined with the result that the PI3K/Akt pathway is activated in UUO kidneys, we tested the Akt inhibitor MK-2206 for its antifibrotic properties and to determine its specific mechanism of action in vivo and in vitro.

MK-2206 is a specific allosteric inhibitor of Akt that acts orally; it inhibits the activation of Akt mainly through binding to the Akt protein in the homologous structural domain of pleckstrin, which results in a conformational change in Akt [30]. MK-2206 is thought to work synergistically with various drugs (AZD6244, venetoclax, etc.) for inhibiting tumor activity, such as in lung cancer and breast cancer. It also enhances the efficacy of the BCG vaccine against Mycobacterium tuberculosis. Based on the genes and pathways changes in the UUO model, we suggest that MK-2206 has promising efficacy in the fibrosis associated with CKD.

In clinical practice, obstructive nephropathy is a common case that may lead to AKI or chronic nephropathy [31]. The UUO model is different from the clinical setting of the obstruction of the unilateral upper urinary tract caused by congenital malformations (UPJO, etc.), stone(s), tumors, inflammation, and iatrogenesis for the reason that it cannot relieve the obstruction. Nevertheless, there is no reliable model of renal fibrosis that relieves the obstruction due to the limitations of experimental techniques. It is reported that, in addition to chronic nephropathy, the UUO model can also be applied to irreversible acute kidney injury [32]. Thus, the renal fibrosis model in this study was constructed using the UUO model, a typical model. The morphological changes and the transformation of collagen fibers in the kidney were observed by H&E and Masson staining, Scr and BUN were used to assess the mice renal function, and then the expression of fibrosis-related indicators was detected by Western blot. Fortunately, the results were as expected. 

Renal inflammation is known to be a major cause of fibrosis. An extensive recruitment and activation of inflammatory cells in the UUO kidney leads to the increased production of cytokines (including pro-fibrotic cytokines, growth factors, and interstitial inflammation) that enhance the inflammatory response [33]. TGF-β1 is a multifunctional cytokine that plays a number of roles in apoptosis, differentiation, cell growth, and trauma repair [34], and is a key regulator of the EMT process, matrix generation, anti-degradation of the matrix, and myofibroblast activation [27,35]. IL-6 also contributes to fibroblast activation and it has been shown that blocking IL-6 ameliorates UUO-induced fibrosis [36]. Our study showed that treatment with MK-2206 significantly slowed the extent and degree of tubular damage in the kidneys of UUO mice and exerted an anti-inflammatory effect by reducing the number of the inflammatory factors TGF-β1, IL-6, and IL-1β.

A variety of injuries target the renal tubular epithelium, and the activation of bioactive mediators in the kidney may contribute to interstitial inflammation and renal fibrosis when the renal tubular epithelium sustains injury [37]. Due to the inhibition of tight junction proteins and other proteins after kidney injury, the interactions between epithelial cells are progressively lost. In turn, epithelial cells acquire a mesenchymal morphology with greater migration capacity and extracellular matrix production, called epithelial–mesenchymal transition (EMT) [38]. EMT is regarded as a key mechanism in renal fibrosis. In the present study, indicators related to EMT were examined in both experimental groups, and E-cadherin expression was found to be downregulated in HK-2 cells induced by TGF-1 and UUO kidney, but partially restored in each group treated with MK-2206, indicating that MK-2206 attenuated the injury of UUO kidney and TGF-β1-induced renal tubular cells. According to studies, the selective removal of Snail and Twist inhibited the development of partial EMT and fibrosis generation in UUO kidneys [39,40]. After MK-2206 treatment, Snail and Twist transcription factors were significantly downregulated compared to UUO and TGF-1 groups, respectively, and it appears that MK-2206 inhibits EMT both in vitro and in vivo.

Myofibroblasts are the main source of ECM. It has been reported that half of the myofibroblasts are derived from fibroblasts in the local environment, while 5% of them are derived from EMT in UUO kidneys [41]. The proliferative transformation of fibroblasts would induce the excessive deposition of ECM in renal fibrosis [42]. Moreover, TGF-β1 plays a key role in the differentiation formation of myofibroblasts in fibrosis [27]. Based on these facts, we consider that MK-2206 can reduce ECM accumulation by suppressing the inflammatory response and decreasing the formation of myofibroblasts. As in other studies, the overexpression of myofibroblasts and ECM was observed in the UUO kidney and TGF-β1-induced HK-2 cells, and the intervention with MK-2206 effectively reduced the expression of myofibroblasts (α-SMA, Vimentin) and extracellular matrix proteins (including Collagen I and Fibronectin). The study showed the potential effects of MK-2206 on inhibiting fibroblast activation (or myofibroblast production) and ECM production, but whether it may also promote ECM degradation is not entirely clear at this time.

Protein kinase B (Akt) has diverse and complex roles in vivo and can phosphorylate a broad spectrum of substrates in the regulation of a wide range of cell and tissue-specific processes. Akt has been studied in a wide range of areas including the regulation of integrin activation, angiogenesis, and ECM expression in fibroblasts [8]. The mammalian target protein of rapamycin (mTOR) is a downriver molecule of Akt, which mediates protein synthesis by regulating the activity of mTOR [43]. mTOR is an atypical serine/threonine kinase that has an extremely significant effect on cell growth, proliferation, survival, metabolism, and senescence [44]. The Akt/mTOR pathway is extensively involved in the fibrosis process in a variety of tissues; for example, it mediated lung fibroblast aerobic glycolysis and collagen synthesis in lipopolysaccharide-induced pulmonary fibrosis [45] and the inhibition of PI3K/Akt/mTOR pathway-activated autophagy, and suppressed peritoneal fibrosis in the process of peritoneal dialysis [46]. For renal fibrosis, it was reported that the Rictor/mTORC2 signaling pathway mediates TGF-β1-induced fibroblast activation [47], and the TGFβ-induced synthesis of the extracellular matrix is mediated by the Akt/mTOR pathway [8]. To further explore the specific mechanism of MK-2206 in alleviating renal fibrosis, we examined the phosphorylation levels of Akt/mTOR pathway-related proteins in vivo and vitro. The results showed that the Akt/mTOR pathway was activated upon the induction of UUO and TGF-β1, which was consistent with our transcriptome sequencing results that showed that the PI3K/Akt pathway was activated, but significantly inhibited upon treatment with MK-2206.

Although previous studies have shown a close relationship between the Akt/mTOR pathway and renal fibrosis, no studies have shown that MK-2206 can be used for the therapy of renal fibrosis. The article confirms that MK-2206 takes effect in alleviating renal fibrosis in animals and cells, and provides new perspectives about the treatment of fibrosis. However, renal fibrosis involves many cells, factors, and pathways, and its process is very complex and variable. The study did not include the influence of MK-2206 on other signaling pathways. Additionally, for the clinical application of drugs, besides their effectiveness, safety also needs to be considered. Therefore, determining whether MK-2206 can be applied to renal fibrosis treatment in humans still requires additional in-depth investigations.

## 5. Conclusions

In conclusion, we demonstrated for the first time the anti-inflammatory and anti-fibrotic effects of MK-2206 in UUO kidney and TGF-β1-induced HK-2 cell fibrosis models in multiple ways, including inhibiting the inflammatory response, EMT, activation of myofibroblasts, and ECM deposition. Mechanistically, MK-2206 acted by inhibiting Akt/mTOR pathway activation. Taken together, these findings indicate that MK-2206 is likely to be a potential therapeutic agent to prevent renal fibrosis.

## Figures and Tables

**Figure 1 cells-11-03505-f001:**
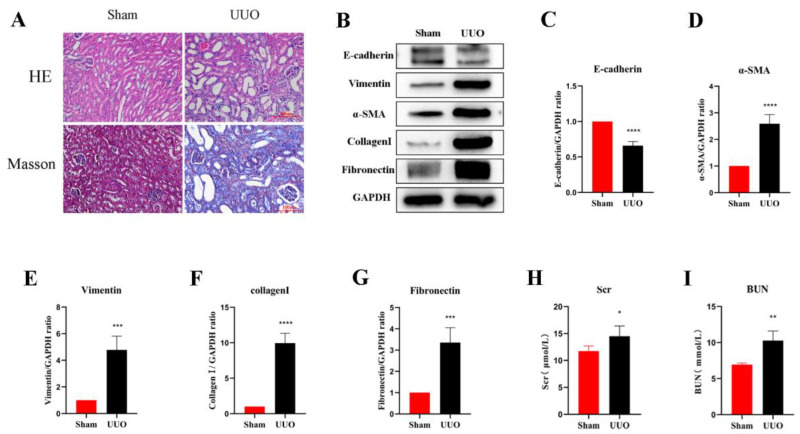
Renal fibrosis models established by UUO. (**A**) Histomorphology (including H&E and Masson staining) of the kidneys in both groups (scale bar: 100 μm). (**B**–**G**) Analysis of E-cadherin, Vimentin, α-SMA, Collagen I, and Fibronectin relative protein levels by Western blot in both groups. (**H**,**I**) Levels of biochemical parameters (Scr and BUN) in both groups. Protein levels were normalized with GADPH. * *p* < 0.05, ** *p* < 0.01, *** *p* < 0.001, **** *p* < 0.0001.

**Figure 2 cells-11-03505-f002:**
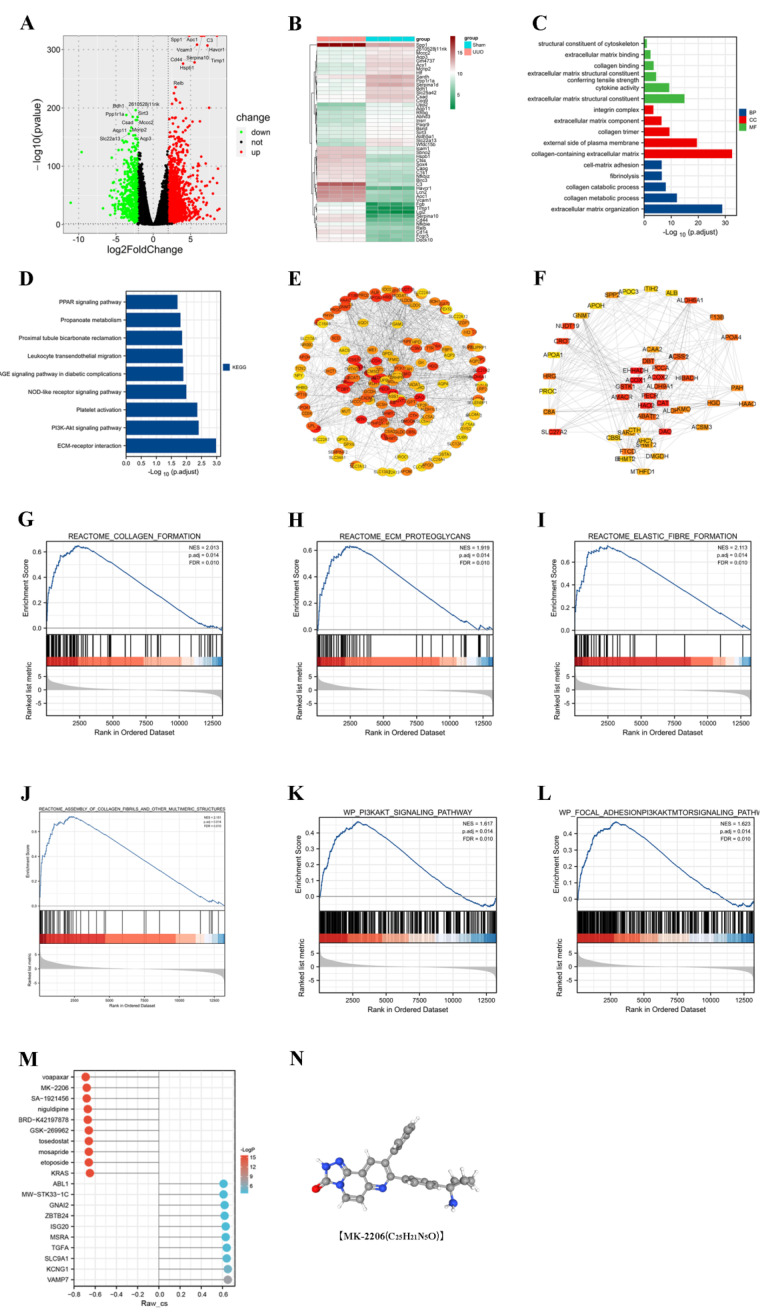
Transcriptome sequencing results analyzed by bioinformatics. (**A**,**B**) Visualization of differentially expressed genes between sham and UUO groups using volcano and heat maps. (**C**,**D**) Gene ontology (GO) and Kyoto Encyclopedia of Genes and Genomes (KEGG) enrichment analysis of differentially expressed genes. (**E**) Construction of protein-protein interaction (PPI) networks to identify and visualize key differentially expressed genes. (**F**) Top 150 selected key differential genes. (**G**–**L**) Exploration of potential pathways using Gene Set Enrichment Analysis (GSEA). (**M**) Prediction of potential anti-fibrotic drugs by the connectivity map (CMap). (**N**) Chemical structure of MK-2206.

**Figure 3 cells-11-03505-f003:**
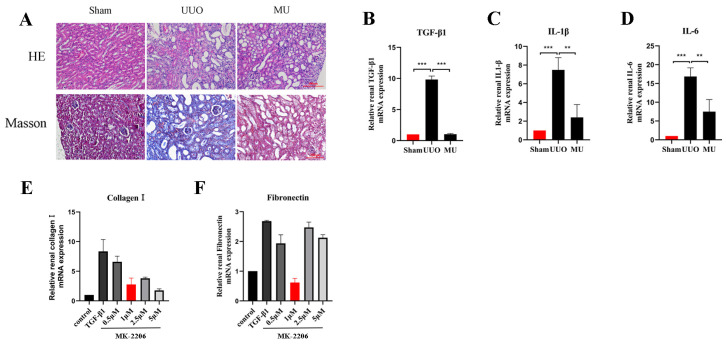
MK-2206 attenuated renal injury and reduced inflammatory factors. (**A**) H&E staining and Masson staining were performed to examine the extent of renal injury and collagen fiber deposition in each group of mice, respectively (scale bar: 100 μm). (**B**–**D**) mRNA levels of inflammatory factors TGF-β1, IL-1β, and IL-6 were detected by real-time PCR in each group of mice. (**E**,**F**) mRNA levels of Collagen I and Fibronectin in HK-2 cells treated by different concentrations of MK-2206 were detected by real-time PCR. The levels of mRNA expression were normalized with GADPH. ** *p* < 0.01, *** *p* < 0.001. MU: UUO + MK-2206.

**Figure 4 cells-11-03505-f004:**
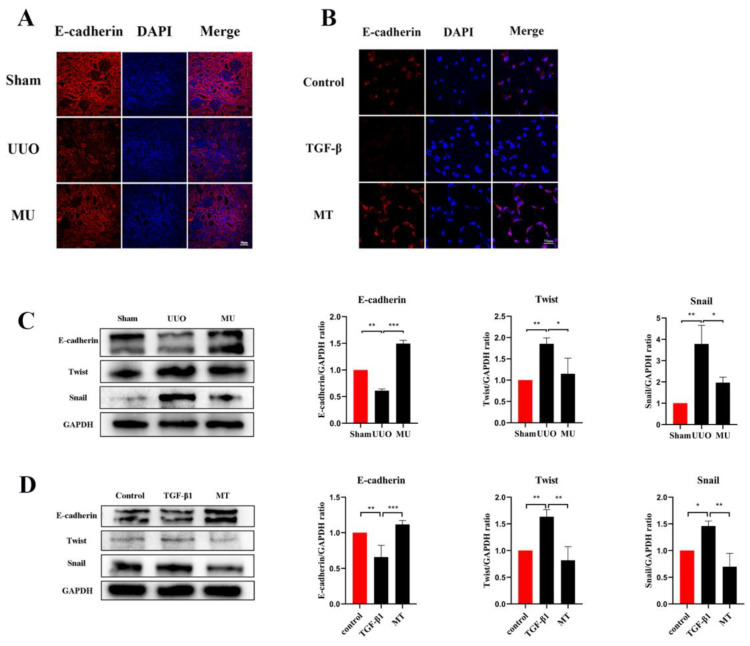
MK-2206 inhibited the EMT process in UUO mice and TGF-β1-induced HK-2 cells. (**A**,**B**) Immunofluorescence staining of E-cadherin in renal tissue and HK-2 cells (scale bar: 50 μm). (**C**,**D**) Western blot analysis of E-cadherin and transcription factors (Snail and Twist). Protein levels were normalized with GADPH. * *p* < 0.05, ** *p* < 0.01, *** *p* < 0.001. MU: UUO + MK-2206, MT: TGF-β1 + MK-2206.

**Figure 5 cells-11-03505-f005:**
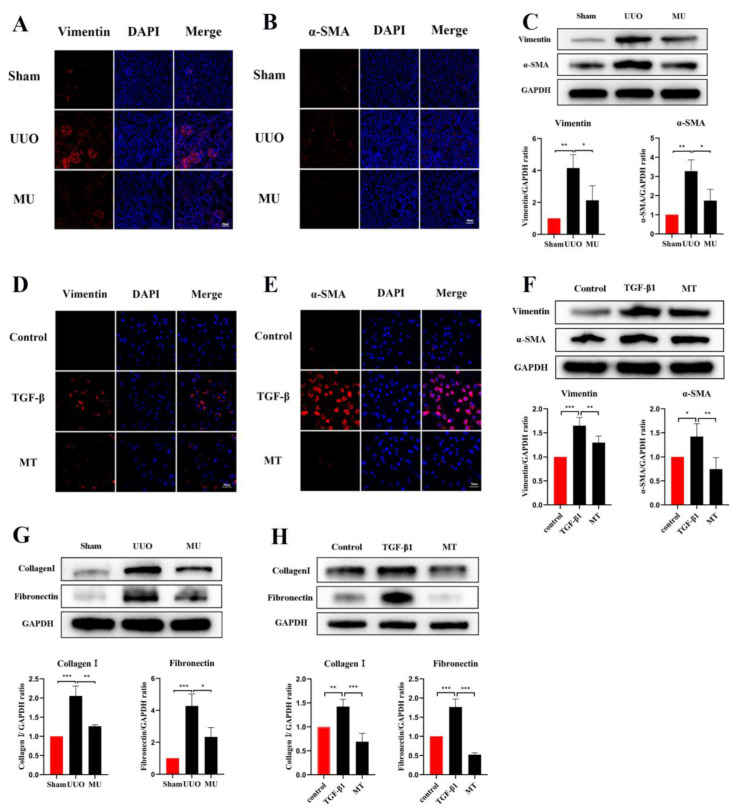
MK-2206 reduced myofibroblast formation and the deposition of extracellular matrix in UUO mice and TGF-β1-induced HK-2 cells. (**A**–**C**) Immunofluorescence staining and Western blot analysis of Vimentin, α-SMA in kidney tissues (scale bar: 100 μm). (**D**–**F**) Immunofluorescence staining and Western blot analysis of Vimentin, α-SMA in HK-2 cells (scale bar: 100 μm). (**G**) Western blot was used to detect the protein expression levels of Collagen I and Fibronectin in the kidney of each group of mice. (**H**) Protein expression levels of Collagen I and Fibronectin in HK-2 cells were detected by Western blot. Protein levels were normalized with GADPH. * *p* < 0.05, ** *p* < 0.01, *** *p* < 0.001. MU: UUO+ MK-2206, MT: TGF-β1 + MK-2206.

**Figure 6 cells-11-03505-f006:**
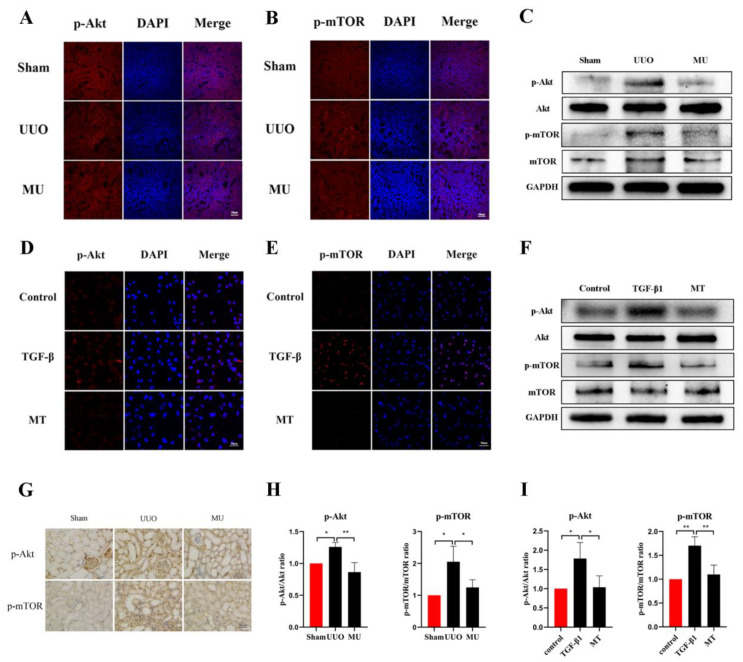
MK-2206 inhibited the activation of the Akt/mTOR signaling pathway in vivo and in vitro. (**A**,**B**) Expression levels of p-Akt and p-mTOR in the kidneys of each group of mice were detected by immunofluorescence staining (scale bar: 50 μm). (**C**,**H**) Protein expression levels of p-Akt and p-mTOR in vivo were detected by Western blot. (**D**,**E**) Expression levels of p-Akt and p-mTOR in HK-2 cells were detected by immunofluorescence staining (scale bar: 50 μm). (**F**,**I**) Western blot was used to detect the protein expression levels of p-Akt and p-mTOR in vitro. (**G**) Immunohistochemical staining of the kidneys in each group of mice using p-Akt and p-mTOR (scale bar: 30 μm). Protein levels were normalized with GADPH. * *p* < 0.05, ** *p* < 0.01. MU: UUO + MK-2206, MT: TGF-β1 + MK-2206.

**Table 1 cells-11-03505-t001:** RT-PCR primer sequences.

Gene	Forward Primer Sequence (5′–3′)	Reverse Primer Sequence (5′–3′)
TGF-β1 (Ms)	ACCGCAACAACGCCATCTATGAG	GGCACTGCTTCCCGAATGTCTG
IL-6 (Ms)	AACCGCTATGAAGTTCCTCTCTG	TGGTATCCTCTGTGAAGTCTCCT
IL-1β (Ms)	CACTACAGGCTCCGAGATGAACAAC	TGTCGTTGCTTGGTTCTCCTTGTAC
Collagen I (H)	GAACAGGGCGACAGAGGCATAAAG	CAACAGGACCAGCATCACCAGTG
Fibronectin (H)	GGCTTGAACCAACCTACGGATGAC	CACCGAGATATTCCTTCTGCCACTG
GAPDH (Ms)	AGGTCGGTGTGAACGGATTTG	TGTAGACCATGTAGTTGAGGTCA
GAPDH (H)	CAAGGCTGTGGGCAAGGTCATC	GTGTCGCTGTTGAAGTCAGAGGAG

Ms: mouse; H: human.

## Data Availability

Not applicable.

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
