# Peer review of "MK-2206 Alleviates Renal Fibrosis by Suppressing the Akt/mTOR Signaling Pathway In Vivo and In Vitro"

_cells, 2022, doi:10.3390/cells11213505_

Round 1
Reviewer 1 Report
1. In figure 3, Authors have mention that ACOX1 and EHHADH two downregulated genes are most critical genes for renal fibrosis. Kindly mention in your result/ discussion part, why this two genes are critical or important.
2. As Author mention, that E-cadherin, a marker of renal tubular epithelial cell in UUO kidney, is there any other major maker available?
Reviewer 2 Report
Renal fibrosis is one of the most serous tissue fibrosis in clinic, second only to the liver, contributes to progressive damage to renal structure and function, leads to significantly worse renal prognosis: chronic kidney disease, or even end-stage renal disease.However, there are no effective drugs for renal fibrosis treatment at present.
In this article, Meiling Chen and colleagues combines transcriptome sequencing and bioinformatics to explore the inhibitory effect of MK-2206 on the occurrence and development of renal fibrosis from the perspective of mice model and cells, and investigates the mechanism of MK-2206 on the pathway of Akt/mTOR, which seed light on the prevention and treatment of renal fibrosis in clinic. In general, this article is of scientific value, relatively complete and innovative, enrich our standing of renal fibrosis with a new therapeutic target. I recommend publish in CELLS. Before that, there are some issues should be clarified.
1.How the time and dose of MK-2206 administration in mice were determined?
2.What was the sample size for transcriptome sequencing and whether there was a statistical difference?
3.How the dosage and timing of TGF-β in in vitro experiments were determined?
4.The mechanism of Akt/mTOR pathway involved in tissue fibrosis should be discussed in more detail, which can help to define the action of MK-2206 on renal fibrosis.
